# Second Surgery for Recurrent Malignant Pleural Mesothelioma after Multimodality Treatment: A Systematic Review

**DOI:** 10.3390/jcm11123340

**Published:** 2022-06-10

**Authors:** Alice Bellini, Sara Mazzarra, Sara Sterrantino, Desideria Argnani, Franco Stella

**Affiliations:** 1Department of Surgery, Division of Thoracic Surgery, G.B. Morgagni—L. Pierantoni Hospital, Via Carlo Forlanini n.34, 47121 Forlì, Italy; sara.mazzarra1@gmail.com (S.M.); sarasterr@gmail.com (S.S.); desideria.argnani@auslromagna.it (D.A.); franco.stella@unibo.it (F.S.); 2Department of Diagnostic and Specialty Medicine—DIMES of the Alma Mater Studiorum, Division of Thoracic Surgery, University of Bologna, G.B. Morgagni—L. Pierantoni Hospital, via Carlo Forlanini n. 34, 47121 Forlì, Italy

**Keywords:** mesothelioma, recurrence, second surgery, post-recurrence survival

## Abstract

Malignant pleural mesothelioma (MPM) is an aggressive asbestos-related tumour with poor prognosis. To date, a multimodality treatment, including chemotherapy and surgery, with or without radiotherapy, is the gold standard therapy for selected patients with epithelioid and early-stage MPM. In this setting, the goal of surgery is to achieve the macroscopic complete resection, obtained by either extrapleural pneumonectomy or pleurectomy/decortication. Failure, in local and/or distant sites, is one of the major concerns; in fact, there has been no established treatment for the recurrence of MPM after the multimodal approach, and the role of surgery in this context is still controversial. By using electronic databases, studies that included recurrent MPM patients who underwent a second surgery were identified. The endpoints included were: a pattern of recurrence, post-recurrence survival (PRS), and the type of second surgery. When available, factors predicting better PRS and perioperative mortality and morbidity were collected. This systematic review offers an overview of the results that are currently obtained in patients undergoing a second surgery for relapsed MPM, with the aim to provide a comprehensive view on this subject that explores if a second surgery leads to an improvement in survival.

## 1. Introduction

Malignant pleural mesothelioma (MPM) is a rare and aggressive malignancy with a poor prognosis, principally caused by a prior asbestos exposure. To date, the multimodal approach, including chemotherapy (CT) and surgery, with or without radiotherapy (RT), with curative intent represents the gold standard therapy for selected patients (early-stage disease with epithelioid histology) [1,2]. For resectable disease, the main surgical procedures consist of extrapleural pneumonectomy (EPP) or pleurectomy/decortication (P/D), performed in order to achieve a macroscopic complete resection [3]. Although there are no randomised comparisons between the two aforementioned techniques, the literature’s data promote the P/D as a less invasive technique with comparable outcomes, but it is still unclear which is better in terms of survival and control of the disease [1,2]. Local and/or distant failure is one of the major issues; in fact, there has been no validated treatment for relapsed MPM after the multimodal protocol and, as far as we known, limited evidence exists concerning the post-recurrence outcomes after the multimodal treatment [4,5,6,7,8,9,10]. In this scenario, the aim of this systematic review is to provide a comprehensive view of this subject that explores if a second surgery leads to improvement in survival.

## 2. Materials and Methods

### 2.1. Literature Search Strategy

A literature search using a formal strategy (PubMed), from January 1980 to December 2021, was performed, using [malignant AND pleural AND mesothelioma] AND [recurrence OR recurrent OR relapse OR second surgery OR redo]. All identified articles were assessed using inclusion and exclusion criteria. The review was accomplished according to PRISMA guidelines [11].

### 2.2. Inclusion and Exclusion Criteria

A PRISMA flow diagram is reported in Figure 1. After the investigation, using the aforementioned keywords, 913 studies were identified. Eligible studies for this review included recurrent MPM patients who underwent a second surgery. The main outcomes considered were the pattern of recurrence, the post-recurrence survival (PRS), and the type of second surgery. When available, the disease-free survival (DFS), the overall survival (OS), the predictors of better PRS, the perioperative mortality and morbidity, the length of hospital stay, and the initial characteristics of recurrent MPM (the type of first surgery, multimodal regimen, sex, age, stage, histology, and side) were also collected. All the selected publications were limited to human subjects and were in the English language. Abstracts, case reports, reviews, comments, editorials, guidelines, and meta-analysis articles were excluded. If not available in the text, pertinent data were extrapolated from tables or figures. Two investigators (A.B. and S.M.) independently reviewed each article. Discrepancies between the two reviewers were resolved by discussion and consensus. The results were reviewed by a senior investigator (F.S.).

## 3. Results

After the selection according to inclusion and exclusion criteria, six studies, published between 2010 and 2021, including a total of 365 relapsed patients, met the inclusion criteria and were included for review. Eighty-nine of them (24.4%) underwent a second surgery. Three studies exclusively analysed patients undergoing EPP [5,12,13] and 1 P/D [8], while two considered patients treated by different surgical approaches (EPP and P/D) [10,13]. All of the studies are retrospective and, among them, five are single-centred [5,8,10,13,14] and one is bicentered [12]. Two studies exclusively analysed patients underwent a second surgery [12,14], while four also included patients who underwent medical treatments or best supportive care [5,8,10,13]. All the information about the study period, the number of relapsed patients, the treatment of the recurrent MPM (surgery, medical treatments, or best supportive care), the type of second surgery, and the pattern of recurrence (local or distant) are reported in Table 1. The DFS, the PRS, the OS, the factors predicting better PRS, the perioperative mortality (30-day) and morbidity (major complications), and the length of hospital stay are represented in Table 2, while Table 3 shows the initial characteristics of recurrent, surgically treated MPM: the type of first surgery (EPP or P/D), the type of multimodal protocol (bimodal or trimodal), the gender, the age at first surgery, the initial stage, the initial histology, and the initial side of the disease. A predominance of local recurrence, chest wall resection, EPP, male gender, and epithelioid subtype is evident.

### 3.1. Basic Characteristics of Recurrent MPM Surgically Treated

The type of first surgical resection was described in all of the studies, with a predominance of the EPP (N = 68, 76.4%). All of the studies except one [13] showed the completion of the multimodal treatment (at least bimodal) for patients who underwent redo-surgery for failure. Three studies [8,10,12] reported information about the multimodal regimen, while the remaining three did not [5,13,14]. Four studies reported the gender and the initial histology [10,12,13,14], with a prevalence of male and epithelioid subtypes.

### 3.2. Pattern of Recurrence and Type of Second Surgery

All the studies reported data about the pattern of the recurrence of patients surgically treated and the type of second surgery. Three studies described surgical resection for both local and distant failure [5,10,12], while three studies described this only for local ones [8,13,14]. Particularly, in the aforementioned studies, 2 (100%) [13] to 47 (100%) [14] patients were surgical treated for local recurrences, while from 0 patients (0%) [8,13,14] to 7 (46.2%) [10] were for distant ones. Globally, surgery was employed with a curative intent, mostly in local relapses (N = 74, 83.2%) rather than in the distant ones (N = 15, 16.8%). The most frequent site of relapse was the chest wall (N = 72, 80.9%), consequently chest wall resection (N = 63, 70.8%) occasionally extended (N = 9, 10.1%) is the commonest intervention performed. Contralateral pulmonary resection, contralateral partial pleurectomy, and ipsilateral single-solid pleural metastasis were employed in three, three, and two cases, respectively. Abdominal surgery was performed in five patients (5.6%), including one case of peritonectomy and hypertermic intraoperative chemotherapy (HIOC).

### 3.3. Post-Recurrence Survival and Factors Predicting Better Outcome after Failure

All studies described the median overall PRS-surgery-related range from 14.5 months [12] to 23.5 months [10], except one reporting the PRS for surgically treated patients, including recurrent MPM patients who underwent medical therapies [8]. Moreover, one study reported PRS separately according to the histologic subtype [14] with much longer expected PRS of patients with the epithelioid subtype. Three studies [5,8,10] also detailed the global PRS (including patients who underwent the best supportive care, besides the treated ones), ranging from 7 [5] to 14.4 [8] months.

All the studies except one [13] analysed factors predicting better PRS; specifically, the DFS [8,10,14], the epithelioid histology [10,14], the local recurrence [5,10], the post-recurrence treatments [5,8,10], and a good Eastern Cooperative Oncology Group performance status (ECOG PS) 0–1 [8] were associated with a prolonged survival. No correlation with the type of first surgery [10] and gender [12] were noted.

## 4. Discussion

Unfortunately, failure after multimodal regimens for MPM is a frequent problem; however, to date there is no validated treatment for the local recurrence of the disease and no recommendations for second surgery for local relapse. The employment of a second surgery with curative intent is still controversial and rarely feasible: as we noted in this systematic review, clinicians chose a second surgery as treatment for the relapse only in 24.4% of the cases, probably due both to the type of the presentation of the relapse itself and the lack of strong evidence about this topic. MPM is formidable and versatile enemy, as demonstrated by its rarity, its inauspicious prognosis, its infinite latency affecting advanced ages, its trend to failure as local dissemination to abdomen and contralateral pleura rather than a single-solid mass. If only one randomised trial exists to date comparing EPP with no surgery in terms of survival and quality of life [15], the realisation of a prospective study in order to verify the usefulness of surgery for relapse is almost improbable for the aforementioned reasons.

In the literature, there are no prospective studies capable of guiding clinicians in the choice of surgery as treatment for recurrence. According to this systematic review, based on retrospective studies, the surgical approach seems to be safely employed in the case of single-solid metastasis [5,8,10,12,13,14].

Burt et al. presented the most numerous case series of patients who underwent a second surgery for a single-solid local failure localised in the chest wall, concluding that the chest wall resection could be an acceptable option for recurrent MPM after MCR in very selected patients with initial epithelioid histology and long DFS [14]. Unfortunately, distant spread tends to present as a dissemination to abdomen and contralateral pleura [16] and rarely occurs with a single-solid pattern. Bellini and collaborators reported the highest number of distant failures surgically treated, all of them presented as a single-solid metastasis, except one [10]. Only a few cases of disseminated distant spread were surgically treated with a partial pleurectomy [5] and peritonectomy associated with HIOC [10]. Nevertheless, Kostron et al. reported a negative experience with a 12.5% rate of major complications ending in death in the case of contralateral partial pleurectomy [5], while no 30-day mortality and major complications were recorded after surgery by Burt et al. and Politi and colleagues for single-solid local [14] and distant [12] relapse, respectively. Hence, contralateral pleurectomy is a high-risk operation that should not be considered for patients with contralateral pleural failure [5]. Moreover, surgery for single-solid recurrence is associated with an acceptable length of hospital stay [12,14].

The surgical resection of the relapse seems to improve the control of the disease [5,10,12,13]: PRS surgery-related, in fact, tends to be longer than global surgery (including patients who received the best supportive care only) [5,8,10]. Across the literature, the post-recurrence treatment is the main predictor of better PRS [5,8,9,10]. A careful selection of the recurrent patients must be carried out by clinicians in order to identify the ones who better could take advantage of a second surgery rather than medical therapies [5,8,10]. The ideal candidate for surgery is a fit patient [8], who underwent multimodal treatment for epithelioid MPM [10,14], presenting a local relapse [5,10] with a long DFS [8,10,14]. No correlation with the type of first surgery is individuate [10]. The predominance of the EPP as first surgery is reported; in fact, only in recent years [17,18], P/D has become the method of choice worldwide.

Local recurrence probably has a less deleterious effect on performance status and, consequently, on survival compared with distant spread [10], while a long DFS perhaps reflects a slower tumour growth speed associated with a less aggressive recurrent disease [7,8,10]. In contrast with the satisfactory PRS after redoing surgery reported by Kostron et al. [5], Bellini and co-workers found tailored medical therapies as the best strategy to face relapse, even in the case of local failure; according to the authors, an early local-only failure may likely reflect a less radical local resection, which could benefit the most from timely systemic therapies rather than more surgery [10].

## 5. Conclusions

This systematic review offers a global overview of the role of a second surgery in patients presenting with the recurrence of MPM after multimodal treatments. The surgical resection seems to be safely employed and associated with good outcomes in the case of a solid single metastasis, preferably if local. A careful selection of patients by a multidisciplinary team is also of paramount importance to maximise benefits. Further studies are needed to better understand this topic. However, the greater centralization of the care of MPM patients is necessary, in order to facilitate scientific reporting; favour inclusion into experimental protocols; and, above all, ameliorate the quality of care for these patients.

## Figures and Tables

**Figure 1 jcm-11-03340-f001:**
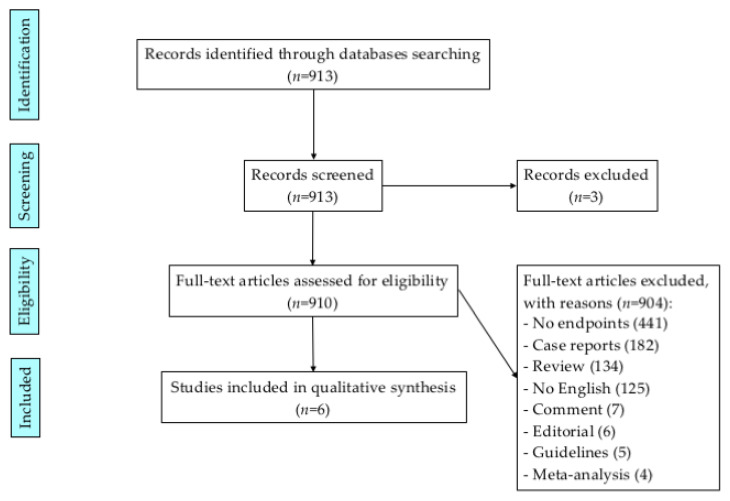
The PRISMA flow diagram.

**Table 1 jcm-11-03340-t001:** The recurrent MPM and relapse treatments.

Author	Study Period	Patients Relapsed, *n* (%)	Relapse Treatment, *n* (%)	Type of Second Surgery, *n* (%)	Pattern of Recurrence of Relapsed MPM Underwent A Second Surgery, *n* (%)
Politi, 2010 [12]	1988–2008, multi-center	53 (93)	Surgery, 8 (15) No surgery, 45 (85)	Chest wall resection, 4 (50) Retroperitoneal resection, 2 (25) Retroperitoneal + pectoral muscle resection, 1 (12.5) Contralateral upper lobe segmentectomy, 1 (12.5)	L, 4 (50) D, 4 (50)
Burt, 2012 [14]	1988–2011, single-center	47 (4.1) with ipsilateral local chest wall recurrence	Surgery, 47 (100)	Chest wall resection, 43 (91.4) Chest wall resection + partial hepatectomy, 2 (4.3) Costophrenic resection extended at the ipsilateral diaphragmatic crura and mediastinum, 2 (4.3)	L, 47 (100)
Okamoto, 2013 [13]	2001–2010, single-center	8 (80)	Surgery, 2 (25) Medical treatment, 6 (75)	Posterior mediastinum resection, 1 (50) Chest wall resection, 1 (50)	L, 2 (100)
Kostron, 2015 [5]	1999–2013, single-center	108 (79), but 106 with complete information	Surgery, 16 (15.1) * Medical treatment, 73 (68.9) * None, 28 (26.4)	Soft tissue chest wall resection, 9 (56.3) Extended chest wall resection, 4 (25) Contralateral partial pleurectomy ± pericardial fenestration ± lung wedge resection and axillary lymphadenectomy, 3 (18.7)	L, 11 (68.7) D, 5 (31.3)
Nakamura, 2020 [8]	2012–2017, single-center	57 (63.3)	Surgery, 3 (5.3) Medical treatment, 40 (70.2) None, 14 (24.5)	Soft tissue chest wall resection, 3 (100)	L, 3 (100)
Bellini, 2021 [10]	1994–2020, single-center	175 (82.5), but 94 with complete information	Surgery, 13 (13.8) ** Medical treatment, 68 (72.3) ** None, 13 (13.8)	Single-solid metastasis resection of Chest wall soft tissue, 3 (23) Ipsilateral pleura, 2 (15.4) Abdomen, 1 (7.7) Contralateral cheek, 1 (7.7) Ipsilateral axillary lymphadenopathy, 1 (7.7) Contralateral pulmonary wedge resection, 2 (15.4) Extended chest wall resection, 1 (7.7) Ipsilateral mastectomy, 1 (7.7) Peritonectomy + HIOC, 1 (7.7)	L, 7 (53.8) ^#^ D, 6 (46.2) ^#^

MPM: malignant pleural mesothelioma; L: local; D: distant; L+D: local+distant; and HIOC: hyperthermic intraperitoneal chemotherapy. * Sixty seven patients (86%) received a single modality treatment, eleven (14%) a combination of two different modalities, and ten (9.4%) surgery alone. ** Fifty-five patients (58.5%) received single modality treatment, twenty-six (27.7%) different combinations, and three (3.2%) surgery alone. ^#^ Unshown data.

**Table 2 jcm-11-03340-t002:** Short- and long-term outcomes of recurrent MPM patients who underwent second surgery.

Author	DFS, Median (Months)	PRS, Median (Months)	OS, Median (Months)	Predictors of Better PRS	30-Day Mortality, N (%)	Major Complications	Length of Stay, Median (Range), Days
Politi, 2010 [12]	NR	Surgery, 14.5	NR	No correlation with Site of recurrence Gender Age at relapse DFS	No	No	10 (8–16)
Burt, 2012 [14]	Surgery, 16.1 Epithelioid 23.4 Biphasic 11.2	Surgery, NR Epithelioid, 20.4 Biphasic, 7.4	Surgery, 44.9 *	Epithelioid histology DFS Age for epithelioid histology	No	No	3 (0–12)
Okamoto, 2013 [13]	All pts, 15.4 Surgery: Patient1, 6 Patient2, 61.7	All pts, 17.8 Surgery: Patient1, 44.6 Patient2, 22.1	All pts, 49.6 * Surgery: Patient1, dead Patient2, alive	NR	NR	NR	NR
Kostron, 2015 [5]	All pts, 9	All pts, 7 Surgery, 16	All pts, 22 **	Local recurrence Second line treatment Redo surgery	2 (12.5)	2 (12.5)	NR
Nakamura, 2020 [8]	All pts, 19	All pts, 14.4 Medical treatments and surgery, 24	All pts, 57 ***	Post-recurrence treatment Performance status 0–1 DFS > 12 months	NR	NR	NR
Bellini, 2021 [10]	All pts, 14 Surgery, 22.8 ^#^	All pts, 12 Surgery, 23.5 ^#^	All pts, 33 * Surgery, 47.2 ^#^	Epithelioid histology Local recurrence DFS ≥ 12 months Post-recurrence medical treatments No correlation with the type of first surgery	No	No	NR

DFS: disease free survival; PRS: post-recurrence survival; OS: overall survival; NR: not reported; EPP: extrapleural pneumonectomy; and P/D: pleurectomy/decortication; pts: patients. * from the first surgery. ** from the first cycle of chemotherapy. *** from the diagnosis. ^#^ Unshown data.

**Table 3 jcm-11-03340-t003:** The basic characteristics of MPM patients who underwent second surgery.

Author	First Surgery, N (%)	Multimodality, N (%)	Sex, N (%)	Age, Median (Range), Years	Pathological Stage According to IMIG TNM8, N (%)	Histology, N (%)
Politi, 2010 [12]	EPP, 8 (100)	Bimodal (surgery + aRT), 8 (100)	Male, 6 (75)	NR	NR	Epithelioid, 8 (100)
Burt, 2012 [14]	EPP, 32 (68) P/D, 15 (32)	NR Global treatments received: iCT,3 (6.4) aCT, 17 (36.2) HIOC, 24 (51.1) aRT, 20 (42.6)	Male, 36 (77)	61.9 (27.3–82.0)	II + IIIA, 10 (21)	Epithelioid, 32 (68) Biphasic, 15 (32)
Okamoto, 2013 [13]	EPP, 2 (100)	NR	Male, 2 (100)	NR	NR	Epithelioid, 1 (50)Biphasic, 1 (50)
Kostron, 2015 [5]	EPP, 16 (100)	NR All pts received at least iCT	NR	NR	NR	NR
Nakamura, 2020 [8]	P/D, 3	Bimodal (surgery + iCT), 3	NR	NR	NR	NR
Bellini, 2021 [10]	EPP, 10 ^#^ P/D, 3 ^#^	Trimodal (surgery + iCT + aRT), 13 ^#^	Male, 12 (92.3) ^#^	64 (54–72) ^#^	IMIG TNM8:^#^ Complete remission, 1 (7.7) I, 11 (84.6) II, 1 (7.7)	Epithelioid, 10 (76.9) ^#^ Biphasic, 2 (15.4) ^#^ Desmoplastic, 1 (7.7) ^#^

EPP: extrapleural pneumonectomy; P/D: pleurectomy/decortication; NR: not reported; pts: patients; aRT: adjuvant radiotherapy; iCT: induction chemotherapy; aCT: adjuvant chemotherapy; and HIOC: hyperthermic intraperitoneal chemotherapy. ^#^ Unshown data.

## Data Availability

Not applicable.

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
