# Peer review of "Second Surgery for Recurrent Malignant Pleural Mesothelioma after Multimodality Treatment: A Systematic Review"

_jcm, 2022, doi:10.3390/jcm11123340_

Round 1

Reviewer 1 Report

Mesothelioma surgery has always been a topic of concern to clinicians. Should it be performed, in which patient, when should it be performed, in what manner, etc.? While EPP was used in earlier years, mesothelioma surgery was switched from EPP to P/D after EPP was defined as "no benefit and possibly harmful" in the MARS study in 2011. However, the number of surgeons still advocating EPP is substantial. On the other hand, information about the second surgery is even more limited, and it is a practice that changes completely depending on the physician and patient. In this sense, this article is valuable in drawing attention to this issue. However, as can be seen, the data in which the publications included patients refer to the years before the study MARS, i.e., the EPP years, so the authors report more on the EPP rate. The question arises: if there were data for more recent years, could these studies be included?
I have one minor criticism: the use of "epithelioid" instead of "epithelial."

Author Response

Dear Reviewer,

We are glad that you appreciated our manuscript. Thank you for your comments that are all valuable and very helpful for revising and improving our paper. Revised portion are marked in red bold in the paper. The main corrections in the paper and the responds to the reviewer’s comments are as following:

If there were data for more recent years, could these studies be included?

We revised published paper until 2021: the only papers that met the inclusion criteria were those discussed in the manuscript, with a total of 89 patients underwent a redo surgery for MPM relapse, 68 after EPP and only 21 after P/D. Three out of the six analysed studies including patients underwent pleurectomy/decortication (P/D), particularly two of them included patients operated after the MARS trial: Nakamura included P/D patients only (study period: 2012-2017) and in my previous paper I reported a series of MPM underwent both EPP and P/D at Padua University Hospital (study period: 1994-2020), the latter mostly after the MARS trial.

The use of epithelioid instead of epithelial.

We changed it in the text.

Reviewer 2 Report

Title: I recommend that the term »radical« would be skipped from the title.

Explanation: In the last few years pleurectomy/decortication is used as the »first« surgery  approach in the frame of a multimodality approach of MP treatment. P/D  is not a »radical« surgery in its basic definition and I do not believe that there is such a thing as a  »second radical surgery« in MPM. Most patients had a metastasectomy as a »redo surgery« .

Abstract: the term radical should be ommited

36: »there are no« instead of there are not

40: limited evidence istead of evidences

43, 54, 56, 71, 75, 78, 86 (Table 1), 93, Table 2, Table 3, 104, 156,192

:  radical sould be ommited

Figure 1: Studies instead of studied

86: Table 1:

-         all of the content is not clearly visible (on the edges), so some information are »missing«

-        # what means »unshown data«?

Table 2:  

OS row: All pts, 57*: from the diagnosis- OS is usually calculated from the date of 1st dignosis until the death of the patient. How was OS defined in other included trils?

»Unshown data: what does that mean?

Table 3: Title: I suggest: Basic chracteristics of MPM patients, who underwent second surgery

Multimodality:

-        Explain bimodal – it is unclear. If bimodal means surgery and systemic treatment and all pts also received RT (ref 12), than is it trimodal ?

-        Explain trimodal

-        After the explanation of the terms, can you maybe put the trials in one of the 3 groups: unimodal, bimodal, trimodal??

Sex: only N and % of male or female is enough, no need to write both (readers can calculate if needed)

Stage:

-is it initial stage?

- stage should be unified /1 staging system) or it is too confusing.  »mediastinal lymph nodes involvement« is not a stage of the disease, this information does not belong there. IMIG TNM 8th: complete remission is not an initial stage of the disease.

Side: does really not matter, this row can be skipped

Unshown: what does that mean?

102: I would focus on patterns of recurrence and  type of second surgery as those were the endpoints of your research. Ommit perioperatove outcomes, as data are scarce and they are already mentioned in the table.

-        add the number and % of pts with local and distant recurrence (data from table 1)

122: PRS according to the histologic subtype (I suggest adding: with expected much longer PRS of patients with epitheloid subtype)

129: 3.3. Chapter 3.3. should be replaced with 3.1. (Basic characteristics should be described before other results)This chapter should be shortened

140: »no vlidate therapy for recurrence« is too broad. It depends on the type of recurrence- if the spread is distat+nt, approach for metastatic disease is recommended.

Maybe it should be stated that there is no validate treatment for local recurrence of the disease and no recommendations for the second surgery for local relapse.

146…its trend to spread as local dissemination to abdomen (not until)

155… who underwent (who missing)

159 … to abdomen

194 carefull instead of accurate

Author Response

Dear Reviewer,

Thank you for your comments concerning our manuscript. Those comments are all valuable and very helpful for revising and improving our paper. We have studied comments carefully and have made correction which we hope meet with approval. Revised portion are marked in red bold in the paper. The main corrections in the paper and the responds to the reviewer’s comments are as following:

Title: I recommend that the term »radical« would be skipped from the title. Explanation: In the last few years pleurectomy/decortication is used as the »first« surgery  approach in the frame of a multimodality approach of MP treatment. P/D  is not a »radical« surgery in its basic definition and I do not believe that there is such a thing as a  »second radical surgery« in MPM. Most patients had a metastasectomy as a »redo surgery«.

We changed it in the text.

Abstract: the term radical should be omitted

We omitted it in the abstract.

36: »there are no« instead of there are not

We changed it in the text.

40: limited evidence istead of evidences

We changed it in the text.

43, 54, 56, 71, 75, 78, 86 (Table 1), 93, Table 2, Table 3, 104, 156,192:  radical sould be omitted

We omitted it in the text.

Figure 1: Studies instead of studied

We changed it in the figure.

86: Table 1:

all of the content is not clearly visible (on the edges), so some information are »missing«

We are sorry, but we do not understand why: if we print the paper all the tables (including the edges) are available in the page.

what means »unshown data«?

Unshown data are data not directly available from my original paper, that I extrapolated using the database utilized for the study.

Table 2:

OS row: All pts, 57*: from the diagnosis- OS is usually calculated from the date of 1st dignosis until the death of the patient. How was OS defined in other included trials?

In the other retrospective studies, the OS was calculated from the date of the first surgery (Burt et al, Okamoto et al, Bellini et al) or from the date of the first cycle of thermotherapy (Kostron et al). We specified it in the caption of the table 2.

Unshown data: what does that mean?

Unshown data are data not directly available from my original paper, that I extrapolated using the database utilized for the study.

Table 3:

Title: I suggest: Basic characteristics of MPM patients, who underwent second surgery

We changed it in the text.

Multimodality:

Explain bimodal – it is unclear. If bimodal means surgery and systemic treatment and all pts also received RT (ref 12), then is it trimodal? Explain trimodal. After the explanation of the terms, can you maybe put the trials in one of the 3 groups: unimodal, bimodal, trimodal??

Bimodal means surgery plus chemotherapy OR radiotherapy, while trimodal means surgery plus chemotherapy AND radiotherapy. In according with this, Politi et al and Nakamura et al are in the bimodal group, while Bellini et al in the trimodal one. While based on the published data it is not possible to classify the others.

We explained it better in the table 3.

Sex: only N and % of male or female is enough, no need to write both (readers can calculate if needed).

We simplified it in the text.

Stage:

is it initial stage?

No, it is not. It is the pathological stage. We specified in the table 3.

stage should be unified /1 staging system) or it is too confusing.  »mediastinal lymph nodes involvement« is not a stage of the disease, this information does not belong there. IMIG TNM 8th: complete remission is not an initial stage of the disease.

All the studies reported the pTNM with different classification or, with the given data, it was not possible to extrapolate the stage of the relapsed patients underwent redo surgery for the recurrence. Based on your important observation, we decided to report it according to the most recent one, i.e. the IMIG TNM8. It was not possible to convert the Butchart one describe by Politi et al so we erased it, while the “mediastinal lymph nodes involvement” (N2 disease) corresponds to the pN1 stage in the IMIG TNM8 staging system, consequently pII and pIIIA stages were added in the table for Burt et al.

Side:

does really not matter, this row can be skipped.

We delete it in the table and in the text.

Unshown: what does that mean?

Unshown data are data not directly available from my original paper, that I extrapolated using the database utilized for the study.

102: I would focus on patterns of recurrence and type of second surgery as those were the endpoints of your research. Omit perioperative outcomes, as data are scarce and they are already mentioned in the table.

We omitted it in the text.

add the number and % of pts with local and distant recurrence (data from table 1)

Those numbers are already mentioned in the text “surgery was employed with a curative intent mostly in local relapses (N=74, 83.2%), rather than in the distant ones (N=15, 16.8%)”, but in a cumulative way. According to your notable observation we specified it in the text, as follow:

“Particularly, in the aforementioned studies 2 (100%) [13] to 47 (100%) [14] patients were surgical treated for local recurrences, while from no patients (0%) [8, 13, 14] to 7 (46.2%) [10] for distant ones”.

122: PRS according to the histologic subtype (I suggest adding: with expected much longer PRS of patients with epithelioid subtype).

This is an important information that we did not well underlined: we added it in the text, as you suggested.

129: 3.3. Chapter 3.3. should be replaced with 3.1. (Basic characteristics should be described before other results). This chapter should be shortened

We put it at the beginning and we shortened it.

140: »no validate therapy for recurrence« is too broad. It depends on the type of recurrence- if the spread is distant, approach for metastatic disease is recommended.

Maybe it should be stated that there is no validate treatment for local recurrence of the disease and no recommendations for the second surgery for local relapse.

Effectively, it is true. Our consideration was too broad. We changed it in the text according to your suggestions.

146…its trend to spread as local dissemination to abdomen (not until)

We changed it in the text.

155… who underwent (who missing)

We changed it in the text.

159 … to abdomen

We changed it in the text.

194 careful instead of accurate

We changed it in the text.
